# A Unified Framework for Deep Symbolic Regression

**Mikel Landajuela**
landajuelala1@llnl.gov

**Chak Shing Lee**
lee1029@llnl.gov

**Jiachen Yang**
yang40@llnl.gov

**Ruben Glatt**
glatt1@llnl.gov

**Claudio Santiago**
prata@llnl.gov

**T. Nathan Mundhenk**
mundhenk1@llnl.gov

**Ignacio Aravena**
aravenasolis1@llnl.gov

**Garrett Mulcahy**
mulcahy4@llnl.gov

**Brenden Petersen**[*]
bp@llnl.gov

Computational Engineering Division
Lawrence Livermore National Laboratory
Livermore, CA 94550

## Abstract

The last few years have witnessed a surge in methods for symbolic regression, from advances in traditional evolutionary approaches to novel deep learning-based systems. Individual works typically focus on advancing the state-of-the-art for one particular class of solution strategies, and there have been few attempts to investigate the benefits of hybridizing or integrating multiple strategies. In this work, we identify five classes of symbolic regression solution strategies—recursive problem simplification, neural-guided search, large-scale pre-training, genetic programming, and linear models—and propose a strategy to hybridize them into a single modular, unified symbolic regression framework. Based on empirical evaluation using SRBench, a new community tool for benchmarking symbolic regression methods, our unified framework achieves state-of-the-art performance in its ability to (1) symbolically recover analytical expressions, (2) fit datasets with high accuracy, and (3) balance accuracy-complexity trade-offs, across 252 ground-truth and black-box benchmark problems, in both noiseless settings and across various noise levels. Finally, we provide practical use case-based guidance for constructing hybrid symbolic regression algorithms, supported by extensive, combinatorial ablation studies.

## 1 Introduction

Symbolic regression (SR) aims to identify a tractable mathematical expression to best fit a dataset. Specifically, given the set of pairs $\{(x_i = (x_{i,1}, \ldots, x_{i,d}), y_i)\}_{i=1}^n$, SR seeks a mathematical function $f : \mathbb{R}^d \to \mathbb{R}$ such that the residual $\|f(X) - y\|^2$ is minimized, where the matrix $X \in \mathbb{R}^{n \times d}$ has entries $X_{i,j} = x_{i,j}$, the vector $y \in \mathbb{R}^n$ has entries $y_i$, and we

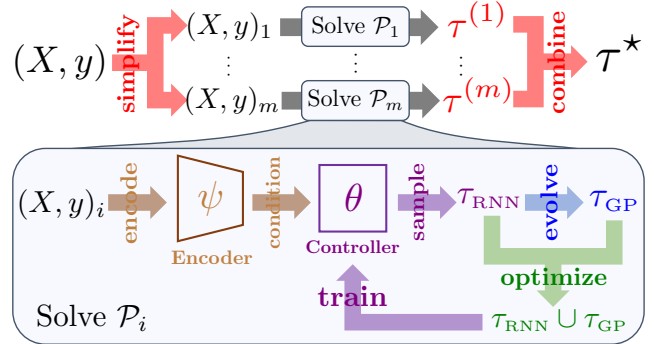

Figure 1: Unified deep symbolic regression. The five integrated solution strategies are color-coded: AIF, DSR, LSPT, GP, LM.

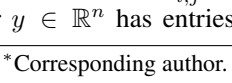

---

[*]Corresponding author.

36th Conference on Neural Information Processing Systems (NeurIPS 2022).

assume point-wise application of real-valued functions over vectors and matrices, i.e., $f(X) = (f(x_i))_{i=1}^n \in \mathbb{R}^n$ (we keep this convention throughout the paper for convenience).

Since the introduction of SR as an application of genetic programming (GP) (Koza, 1994), many classes of methods—especially machine learning-based methods in recent years—have been proposed. These methods include recursive problem simplification (Udrescu et al., 2020), neural-guided search (Petersen et al., 2021a), large-scale pre-training for problem generalization (Biggio et al., 2021; Kamienny et al., 2022; Vastl et al., 2022), sparse linear regression using nonlinear basis functions (Brunton et al., 2016), and combination of evolutionary search with learning (Mundhenk et al., 2021). Each of these independent advances exploits different information (e.g., graph modularity or constraints) to offer key capabilities for various aspects of the SR problem (e.g., search or generalization), but the specificity of each work resulted in design choices that pose limitations in other facets of the problem. This motivates an investigation of whether disparate solution strategies are complementary to one another and how they can be effectively combined. In this paper, we carefully rewire the five aforementioned solution strategies into a unified SR framework and show that they are indeed complementary.

Based on the SRBench pipeline for benchmarking SR algorithms (`https://github.com/cavalab/srbench`), our unified deep symbolic regression (uDSR) framework achieves a new state-of-the-art in SR: for ground-truth problems, uDSR ranks highest in both symbolic-based and accuracy-based solution rates for both noiseless settings and for multiple noise levels; for black-box problems, uDSR falls on the accuracy-complexity Pareto frontier alongside three other non-dominated algorithms.

Our main contributions in this work include: (1) a modular, unified framework for SR that integrates five disparate SR solution strategies (illustrated in Figure 1); (2) a novel LINEAR token that identifies sparse coefficients of a linear combination of basis functions and can be used by almost any SR algorithm; (3) achieving new state-of-the-art for SR on SRBench in its metrics for symbolic recovery, solution accuracy, and accuracy vs complexity trade-offs; and (4) practical recommendations for designing SR algorithms, driven by extensive and combinatorial ablation studies that show the complementary benefits of these previously disjoint SR solution strategies.

## 2   Background

We describe each of the five solution strategies our framework unifies. For each strategy, we focus on a *key capability* and a *key limitation*[2], summarized in Table 1, which motivates our unification that captures all capabilities and mitigates all limitations via the inherent synergy among strategies. We encourage readers to read the original works for additional background.

**AI Feynman.** AI Feynman 2.0 (AIF) (Udrescu et al., 2020) approaches SR via *recursive problem simplification*; it is the only known algorithm to date to employ this solution strategy. Specifically, AIF exploits knowledge of physics and the given training data, $(X, y)$, by identifying simplifying properties (e.g., multiplicative separability) of the underlying functional form. Applying a simplification yields sub-problems (or one sub-problem) of lower dimensionality. Simplifications are applied recursively until all sub-problems (i.e., leaf nodes of the recursion tree) are 1-dimensional. For each sub-problem, the paper employs a combination of brute force search plus polynomial fit (BF/PF). Finally, solutions to the sub-problems are then combined back into a final solution to the original problem. While AIF positions itself as a standalone SR algorithm, it is more appropriately understood as a *SR problem simplification algorithm*, as pointed out in prior work (Petersen et al., 2021a; Biggio et al., 2021). From this general perspective, the use of BF/PF is simply a design choice, orthogonal to the core methodology, and may be enhanced by integration with more advanced search methods for solving sub-problems.

**Deep symbolic regression.** Deep symbolic regression (DSR) (Petersen et al., 2021a) leverages the effectiveness of deep reinforcement learning for combinatorial search in SR. DSR employs a neural network *controller* with parameters $\theta$ to represent a distribution $p(\tau|\theta)$ over mathematical expressions $\tau = [\tau_1, \ldots, \tau_T]$, represented by the sequence of "tokens" $\tau_i$ (mathematical operators, input variables, or constants) in the pre-order (depth-first then left-to-right) traversal of the corresponding *expression*

---

[2]We clarify that the key limitations are not necessarily issues with the algorithms themselves; often, they are conscious design decisions by the authors. For example, in the context of physics-based SR problems, AIF's use of BF/PF only strengthens their claim that recursive simplification alone is often sufficient to solve the problem. This work considers a wider context of challenging SR problems, in which these can be viewed as limitations.

*tree* (see Koza (1994) for details). For instance, $\tau = [+, \text{SIN}, x_1, \text{EXP}, x_2]$ corresponds to expression $f(x) = \sin(x_1) + \exp(x_2)$. Expressions are sampled from the controller autoregressively, i.e. $p(\tau|\theta) = \prod_{i=1}^{T} p(\tau_i|\tau_{1:i-1}, \theta)$, and evaluated based on fitness $R(\tau, (X, y))$ to the $(X, y)$ dataset. The controller is then trained to maximize the reinforcement learning objective:

$$J(\theta) := \mathbb{E}_{\tau \sim p(\tau|\theta)} \left[ R(\tau, (X, y)) \right], \tag{1}$$

using a risk-seeking policy gradient that aims to maximize best-case performance. Autoregressive sampling affords the ability to easily incorporate prior knowledge by directly modifying likelihoods of individual tokens $p(\tau_i|\tau_{1:i-1}, \theta)$ *in situ*, for example by precluding expressions with nested trigonometric operators. This dramatically reduces the search space, which has been shown to greatly improve the ability to recover ground-truth expressions (Petersen et al., 2021b). Whereas DSR focuses on discrete search and pruning the combinatorial search space, it does not exploit the $(X, y)$ data. In fact, the $(X, y)$ data itself only shows up in the reward computation in (1), which is a weak, aggregate learning signal. Thus, DSR trains *from scratch* for each new SR problem and cannot generalize to new problems.

**Large-scale pre-training.** Large-scale pre-training (LSPT) algorithms (Biggio et al., 2021; Kamienny et al., 2022; Vastl et al., 2022) tackle the problem of *generalization* in SR: they seek a *single* model that, once trained, produces a distribution over solutions to *any* SR problem. To accomplish this, the algorithm "Neural Symbolic Regression that Scales" (NeSymReS) (Biggio et al., 2021) uses a set transformer (Lee et al., 2019) to learn an encoding of the $(X, y)$ data. A second transformer network then maps the encoding to a distribution over expressions using autoregressive sampling. The model is trained end-to-end via supervised learning, using a large dataset of SR problems (input data $(X, y)$ and corresponding ground truth label $\tau$) produced by a programmatic expression generator (Lample and Charton, 2019). Given $(X, y)$ at test time, beam search is used to find high-likelihood expressions. LSPT approaches claim that pre-training alone is sufficient to solve many SR problems. However, the lack of fine-tuning during testing means that LSPT algorithms may not be optimized for new out-of-distribution SR problems.

**Genetic programming.** Genetic programming (GP) refers to a well-established class of evolutionary algorithms commonly applied to SR in which a set or "population" of mathematical expressions is maintained and "evolved" over time using genetic operators such as mutation, crossover, and selection. We refer the reader to Langdon and Poli (2013) for an overview of GP. In this work, we consider only standard GP (as defined in Chapter 7 of Bäck et al. (2018)); however, our integration of GP into the unified framework applies to virtually any GP method. GP excels at exploring the search space to find complex expressions, but does not employ a neural network or other parameterized model to *learn* over time. This limitation was addressed in Mundhenk et al. (2021), where GP was combined with a learned expression generator, but not in concert with other methods considered here.

**Linear models.** Linear models (LM) are a mainstay of statistics and machine learning (Murphy, 2022). LM methods seek expressions of the form $g = \sum_{i=1}^{k} \beta_i \phi_i = \Phi \beta^{\mathsf{T}}$, where $\Phi := (\phi_1, \dots, \phi_k)$ comprises a library of $k$ user-defined basis functions $\phi_i : \mathbb{R}^n \to \mathbb{R}$, and $\beta = (\beta_1, \dots, \beta_k)$ is a vector of scalar coefficients that is often learned using least squares minimization. To avoid over-fitting, the vector $\beta$ can be alternatively optimized using sparsity inducing optimizers. As an example, the algorithm "Sparse Identification of Nonlinear Dynamical systems" (SINDy) (Brunton et al., 2016) is a simple, computationally expedient, and popular LM method for SR, particularly for learning systems of differential equations. SINDy identifies sparse representations by using LASSO (Tibshirani, 1996). While effective at quickly learning sparse expressions, the restriction to a linear combination of pre-defined basis functions results in a massively reduced search space, precluding such functional forms as rationals and compositions. In our unified framework, we abstract the entire process of learning a linear model into one single discrete token, thereby restoring the nonlinear, combinatorial search space while retaining the ability to quickly learn sparse linear subcomponents.

## 3 Related Work

In addition to AIF, DSR, LSPT, GP, and LM described above, we describe several other related SR solution strategies that are not part of our unified framework. EQL$^{\div}$ presents a unique deep-learning framework for SR that employs neural networks whose activation functions are elementary operators, enabling end-to-end differentiable training (Sahoo et al., 2018). GrammarVAE learns to encode expression trees in a continuous space and uses Bayesian optimization to optimize them in latent

Table 1: Key capabilities and limitations of the five unified solution strategies.

| Solution strategy | Label | Key capability | Key limitation |
|---|---|---|---|
| AI Feynman | AIF | Exploits $(X, y)$ data to simplify a SR problem into lower-dimensional sub-problems. | Underlying SR algorithm resorts to simple BF/PF. |
| Deep symbolic regression | DSR | Neural network learns over time, with the ability to incorporate *in situ* constraints. | Does not exploit the $(X, y)$ data, which only appears in the (weak) reward function. |
| Large-scale pre-training | LSPT | Leverages big data, learning from many other problems by conditioning on the $(X, y)$ data. | Limited to fixed numbers of dimensions, and does not fine-tune the search to a particular SR problem. |
| Genetic programming | GP | Rapidly explores the search space via genetic operators. | Does not employ a neural network to learn over time. |
| Linear models | LM | Quickly learn sparse coefficients of a linear combination of basis functions. | Operate on a dramatically reduced search space. |

space (Kusner et al., 2017). Both EQL$^{\div}$ and GrammarVAE impose specific representations of the expressions (either by constraining the search space or by projecting it into a latent space) and thus do not facilitate straightforward integration with the solution strategies in our framework. The idea of pre-training models for use in neural-guided search, used by LSPT approaches and our unified framework, is inspired by Bello et al. (2017), who apply a similar method to the traveling salesman problem.

Previous work on hybrid approaches to SR mainly focus on improving a single aspect of one approach, such as alleviating the challenge of high dimensional data by extracting features for GP (Icke and Bongard, 2013), or improving the starting population of GP iterations (Mundhenk et al., 2021). Often, the motivation for combining separate methods comes from specific problem settings, such as the case of datasets with missing values (Al-Helali et al., 2018) or noisy high-dimensional measurements in systems with known physical constraints (Reinbold et al., 2021). In contrast, the unification in our work serves to enhance multiple general capabilities of SR, such as generalization, combinatorial search and optimization, and single-token abstraction of function spaces.

Our empirical analysis heavily leverages SRBench, a reproducible and open-source platform for benchmarking new SR methods against 14 established methods on accuracy and complexity for 122 real-world datasets and exact symbolic correctness for 130 ground-truth problems with varying noise levels (La Cava et al., 2021). SRBench has seen rapid adoption by recent works that use it to evaluate improvements in accuracy, exact solution rate, and solution complexity (Kamienny et al., 2022; Virgolin and Bosman, 2022; Zhang et al., 2022).

Lastly, we note a similarity between this work and Rainbow, a work that combines several deep reinforcement learning methods into a unified algorithm and performs extensive ablation studies (Hessel et al., 2018). Analogous to Rainbow's use of Deep Q Networks (DQN) as a base framework into which it integrates several improvements, this work uses DSR as a base framework into which we integrate AIF, LSPT, GP, and LM.

## 4 Methods

Our overall unification strategy is to carefully rewire the solution strategies identified above as connected but non-overlapping modules in an algorithmic framework that leverages their key capabilities. This is visualized in Figure 1 (see also Figure 12 in Appendix L for an illustrative example of expression discovery under uDSR). In an initial offline stage, pre-training a parametric controller model on a large dataset facilitates generalization to test problem instances (LSPT). In the online stage, given a specific problem instance, a recursive problem simplification module (AIF) exploits modularity of mathematical expressions to produce sub-problems of lower dimensionality for the main trunk of uDSR. Within the trunk, a Recurrent Neural Network (RNN) continually learns over many iterations (DSR) and provides good candidates to seed the starting population for genetic programming (GP); high fitness populations produced by GP are combined with elite candidates for

controller parameter updates via risk-seeking policy gradient. Finally, the RNN is permitted the use of a powerful symbolic token that encapsulates an entire space spanned by basis functions (LM).

Below, we describe the integration of each module in detail. Algorithm 1 provides high-level pseudocode for uDSR. More detailed pseudocode is available in Algorithm 2 of Appendix A.

Table 2: Feature compatibility matrix for uDSR and the five integrated solution strategies.

| | Recursive problem simplification | Search space pruning | Leverages big data | Evolutionary search | Linear constant optimization | Nonlinear constant optimization | Neural-guided search |
|------|:---:|:---:|:---:|:---:|:---:|:---:|:---:|
| AIF | ✓ | | | | | | |
| DSR | | ✓ | | | | ✓ | ✓ |
| LSPT | | | ✓ | | | ✓ | ✓ |
| GP | | | | ✓ | | ✓ | |
| LM | | | | | ✓ | | |
| uDSR | ✓ | ✓ | ✓ | ✓ | ✓ | ✓ | ✓ |

**Beginning with DSR.** For DSR, we begin with the method proposed by Petersen et al. (2021a), including two improvements to exploration (soft length prior and hierarchical entropy regularizer) introduced in follow-up works (Landajuela et al., 2021a,b). DSR includes a number of *in situ* priors and constraints to the search space, which help bias and/or prune the search, and have been shown to greatly improve performance (Landajuela et al., 2021b). We provide a complete list of DSR priors and constraints in Appendix B, including several new constraints required to unify all five methods (described below). DSR also includes a CONST token that represents an arbitrary floating-point constant, whose value is optimized using a nonlinear optimizer, e.g., L-BFGS-B (Zhu et al., 1997).

**Integrating AIF via recursive simplifications.** Our key insight to integrate AIF (Udrescu et al., 2020) is that its recursive problem simplification steps can be viewed as orthogonal to the underlying SR algorithm. While AIF resorts to BF/PF, we propose using a hybrid of the other four integrated algorithms (which we continue to describe in subsequent paragraphs). To integrate AIF, we consider the starting SR problem as $\mathcal{P}_1$, a $d$-dimensional SR problem. One application of AIF's simplification step produces either one sub-problem $\mathcal{P}_2$, with dimensionality $d_2 < d$, or two sub-problems, $\mathcal{P}_2$ and $\mathcal{P}_3$, with dimensions $d_2$ and $d_3$ such that $d_2 + d_3 = d$, as well as metadata $\mathcal{M}$ containing information required to recombine sub-solutions back to the full solution of $\mathcal{P}_1$. Recursive simplifications are applied until sub-problems at the leaf nodes of the recursion tree are each 1-dimensional. For each sub-problem $\mathcal{P}_i$ (including the starting problem, $\mathcal{P}_1$), we apply our hybrid SR algorithm to its corresponding dataset, $(X, y)_i$. Finally, using the metadata from each simplification step, solutions from all sub-problems are recombined to form a single, final solution to the original problem, $\mathcal{P}_1$.

**Integrating LM via the novel LINEAR token.** Our design goal for integrating LM is to retain the ability to quickly learn linear subcomponents of an expression, while still operating in a general nonlinear functional space. That is, while LM finds expressions of the form $g(x) = \Phi(x)\beta^\mathsf{T}$, we seek to search more generic expressions of the form $f(x) = F(x, g(x))$, where $F$ is a nonlinear function subject to certain constraints (see below), while still using linear methods to determine the coefficients of $g(x)$. We accomplish this by proposing to represent $g(x)$ as a special type of terminal token, which we call LINEAR. LINEAR serves as a placeholder for a LM-like solution, $\Phi(x)\beta^\mathsf{T}$, where the values of the coefficients $\beta$ are determined only when a traversal has terminated. Thus, we abstract the entire process of learning a linear model into a single discrete token. In our approach, the function $F(x, w)$ must admit an expression tree with a single argument token for $w$ and there must exist $G(x, v)$ such that $y = F(x, G(x, y)), \ \forall x, y \in \mathbb{R}^d \times \mathbb{R}$.

Given an expression $f(x) = F(x, \Phi(x)\beta^\mathsf{T})$, with $F$ as described above, let $\overline{f}(\Phi\beta^\mathsf{T})$ denote the view of $f$ as a function of $\beta$ only. Naively, one could solve a nonlinear optimization problem to determine the values of $\beta$, just as DSR and Biggio et al. (2021) optimize CONST tokens. However, nonlinear optimization is computationally expensive, scales poorly with more coefficients, and is challenging to induce sparsity. Instead, we consider the following *convex* optimization problem, for which there

are efficient sparse linear solvers:

$$\operatorname*{arg\,min}_{\beta \in \mathbb{R}^k} L(\beta) := \|\Phi(X)\beta^\mathsf{T} - \overline{f}^{-1}(y)\|_2^2. \tag{2}$$

We attain this formulation by solving the equation $y = F(X, \text{LINEAR})$ for LINEAR, which gives LINEAR $= G(X, y)$. Since this requires taking the inverse of all ancestor operators of LINEAR—binary operators are reduced to unary operators by lexical closure, elaborated more programmatically below—we use $\overline{f}^{-1}(y) := G(X, y)$ to denote the view of $G$ as a function of $y$ only.

Equation (2) exhibits an attractive property that enables LINEAR to exactly recover non-linear expressions with linear subcomponents. When LINEAR is chosen at the correct position in a traversal, in the sense that the true computational graph contains a linear combination of basis functions in the corresponding subgraph, we have that $\overline{f}^{-1}(y)$ exactly equals that linear combination. For example, given $(X, y)$ data generated from a ground-truth expression $f^\star(x) = \sin(x_1) + \exp(1.2x_1^3 + 3.4x_1x_2)$, and a basis containing all monomials up to degree 3, the traversal $\tau = [\text{+}, \text{SIN}, x_1, \text{EXP}, \text{LINEAR}]$ yields $\overline{f}^{-1}(y) = \log(y - \sin(x_1))$, and the corresponding solution to (2) *exactly* recovers $f^\star$.

Programmatically, we solve (2) using the expression tree for $\tau$ and the following procedure (illustrated in Figure 5 of Appendix C; pseudocode is provided in Algorithm 2 of Appendix A): First, partially execute the tree by evaluating all nodes that are not direct ancestors of LINEAR. Second, redefine binary operators as unary operators by fixing (or "binding") the one pre-computed argument (this is called *partial application*—a type of *lexical closure*—in computer science). Third, successively apply the root node's inverse operation on the $y$ data until LINEAR is the only token in the tree, resulting in $\overline{f}^{-1}(y)$. Finally, perform a linear fit on (2) using the transformed data, $(X, \overline{f}^{-1}(y))$.

This procedure for abstracting a linear model into a single operator exhibits several complications: (1) it only works if there is exactly one LINEAR token, (2) it does not work in the presence of other placeholder tokens, e.g., the CONST token used in DSR, and (3) it does not work if LINEAR has non-invertible ancestors (e.g., SIN), as $y = F(X, \text{LINEAR})$ might not have a unique solution on LINEAR. Fortunately, seamlessly incorporating these restrictions into the search space is precisely what the DSR algorithm excels at. To this end, we enable LINEAR into our framework by introducing three constraints to the search space (further detailed in Appendix B): (1) there can be at most one LINEAR token; (2) the LINEAR token and CONST token are mutually exclusive—one cannot appear if the other is present; and (3) the LINEAR token cannot be the descendant of a non-invertible unary token, e.g., SIN. In this work, we consider the set of basis functions $\Phi$ given by the monomials of degree up to $D$ and solve (2) using a sparsity-inducing least square method (see Appendix C for details).

Lastly, we observe that standard LM regression is restored when $F(x, w) = w$. In that case, we recover the function $f(x) = \Phi(x)\beta^\mathsf{T}$ with trivial traversal $\tau = [\text{LINEAR}]$. Thus, our method expands the search space of LM by allowing nonlinear function compositions. For instance, $F(x, w) = \frac{1}{w}$ leads to $f(x) = \frac{1}{\Phi(x)\beta^\mathsf{T}}$, which is outside the search space in standard LM regression.

**Integrating GP via population seeding.** To our knowledge, hybridization of DSR and GP is the only combination of the five methods that has been explored to date. To summarize, as in Mundhenk et al. (2021), we hybridize DSR with GP by abstracting the GP algorithm as an inner optimization loop within DSR. Each time DSR samples a batch of expressions, we use this batch as the starting population for an inner-loop GP algorithm. The top samples at the end of GP are combined with the DSR batch for training the neural network. This process leverages GP's ability to rapidly explore the search space, while using DSR's persistent neural network to mitigate the disadvantages of GP's non-parametric nature (Mundhenk et al., 2021).

**Integrating LSPT via supervised or reinforcement learning.** We integrate LSPT by merging the RNN used in DSR with a set transformer in an encoder-decoder fashion. Our encoder architecture closely mimics that of NeSymReS (Biggio et al., 2021). The set transformer encodes a given dataset $(X, y)$ into a latent representation $h_0$. This representation is passed as initial state to the RNN, which decodes it into an expression $\tau$. The resulting architecture models the distribution $p(\tau|\theta, \psi, (X, y))$, where $\theta$ and $\psi$ are the parameters of the RNN and set transformer, respectively (see Figure 1). As in Biggio et al. (2021), we pre-train the model end-to-end on many SR problems using supervised learning, using ground truth expressions as labels, resulting in parameters $\theta^\star, \psi^\star$. However, different from existing LSPT approaches, given a new dataset $(X', y')$, we propose to search for a symbolic

---

**Algorithm 1** Unified deep symbolic regression. Color key: AIF, LSPT, DSR, GP, LM

---

**input** Symbolic regression problem $\mathcal{P}$ consisting of tabular data $(X, y)$
**output** Best fitting expression $\tau^\star$
**parameters** Pre-trained controller and encoder parameters $(\theta^\star, \psi^\star)$, objective function $J(\theta, \psi)$
1: $\mathcal{P}_1, \ldots, \mathcal{P}_m \leftarrow$ use AIF to recursively simplify $\mathcal{P}$ into $m$ sub-problems
2: **for** each sub-problem $\mathcal{P}_i$ and corresponding dataset $(X, y)_i$ **do**
3:    $\theta = \theta^\star, \psi^\star \leftarrow$ load pre-trained encoder and controller parameters
4:    **while** budget not exceeded **do**
5:       $\mathcal{T}_{\text{RNN}} \leftarrow$ sample $N$ expressions from $p(\tau|\theta, \psi^\star, (X, y)_i)$
6:       $\mathcal{T}_{\text{GP}} \leftarrow$ evolve population by performing $S$ generations of GP
7:       $\mathcal{T} \leftarrow$ combine $\mathcal{T}_{\text{RNN}}$ and top $k$ expressions from $\mathcal{T}_{\text{GP}}$
8:       $\mathcal{T} \leftarrow$ compute coefficients of LINEAR tokens
9:       $\theta \leftarrow$ train controller on $\mathcal{T}$ according to $J(\theta, \psi^\star)$
10:    $\tau^{(i)} \leftarrow$ store best expression for problem $\mathcal{P}_i$
11: $\tau^\star \leftarrow$ use AIF to combine solutions $\tau^{(1)}, \ldots, \tau^{(m)}$ into final solution
12: **return** $\tau^\star$

---

expression by *fine-tuning* $p(\tau|\theta, \psi^\star, (X', y'))$ using reinforcement learning, with initial conditions $\theta_0 = \theta^\star$.

The pre-training phase requires a large dataset comprising SR problems, each defined by a set of points $(X, y)$ along with true expression $\tau$. To scale to arbitrary dataset size, we seek a problem instance generator $p_\mathcal{G}(\tau, (X, y)) = p_\mathcal{G}((X, y)|\tau)p_\mathcal{G}(\tau)$, from which we can sample expressions $\tau$ and corresponding $(X, y)$ data. Full details of the sampling process used by $p_\mathcal{G}$ are provided in Appendix D. Briefly, we leverage the fact that the distribution induced by DSR includes a prior (Petersen et al., 2021b). To generate expressions, $p_\mathcal{G}(\tau) = p(\tau|\theta = \mathbf{0})$ directly samples from this prior, ignoring emissions from the RNN. Thus, the algorithm bootstraps or "imagines" expressions that it then uses for training data, rather than relying on an external expression generator as in Biggio et al. (2021); Kamienny et al. (2022); Vastl et al. (2022). After sampling $\tau$, $p_\mathcal{G}((X, y)|\tau)$ is obtained following the strategy in Biggio et al. (2021).

In this work, we explore two different objectives for model pre-training. First, following Biggio et al. (2021), we consider the supervised learning (SL) objective:

$$J_{\text{SL}}(\theta, \psi) := \mathbb{E}_{\tau_g, (X, y)_g \sim p_\mathcal{G}} \left[ \log p\left(\tau_g | \theta, \psi, (X, y)_g\right) \right].$$

During fine-tuning, ground-truth labels will not be available. To mitigate an objective mismatch between pre-training and fine-tuning objectives, we alternatively explore pre-training *à la* reinforcement learning (RL) (Bello et al., 2017). Specifically, we consider:

$$J_{\text{RL}}(\theta, \psi) := \mathbb{E}_{(X, y)_g \sim p_\mathcal{G}} \left[ \mathbb{E}_{\tau \sim p(\tau|\theta, \psi, (X, y)_g)} \left[ R(\tau, (X, y)_g) \right] \right]. \tag{3}$$

In practice, this objective is maximized using a novel formulation of the risk-seeking policy gradient for multitask learning (see Appendix D for details).

## 5 Experiments

**Pre-training.** We pre-train four models: two trained using SL and two trained using RL, each with and without the LINEAR token. Details of the pre-training setup can be found in Appendix D.

**Experimental setup using SRBench.** We empirically assess uDSR using SRBench (La Cava et al., 2021), an open-source and reproducible pipeline for benchmarking SR algorithms. SRBench features 130 problems with hidden ground-truth analytic solutions and 122 real-world datasets with no known analytic model ("black-box" problems) from the PMLB database (Olson et al., 2017). By running uDSR through the SRBench pipeline, we enable direct comparison to its curated results of 14 contemporary SR methods. The 14 baselines are described in Appendix E.

For all experiments, we use a minimal set of tokens: $+, -, \times, \div$, SIN, COS, EXP, LOG, SQRT, 1.0, CONST, and (except for the appropriate ablations) LINEAR. For simplicity, our choice of $\Phi$ (basis functions for LINEAR) includes only polynomial terms up to degree 3. Hyperparameters are listed

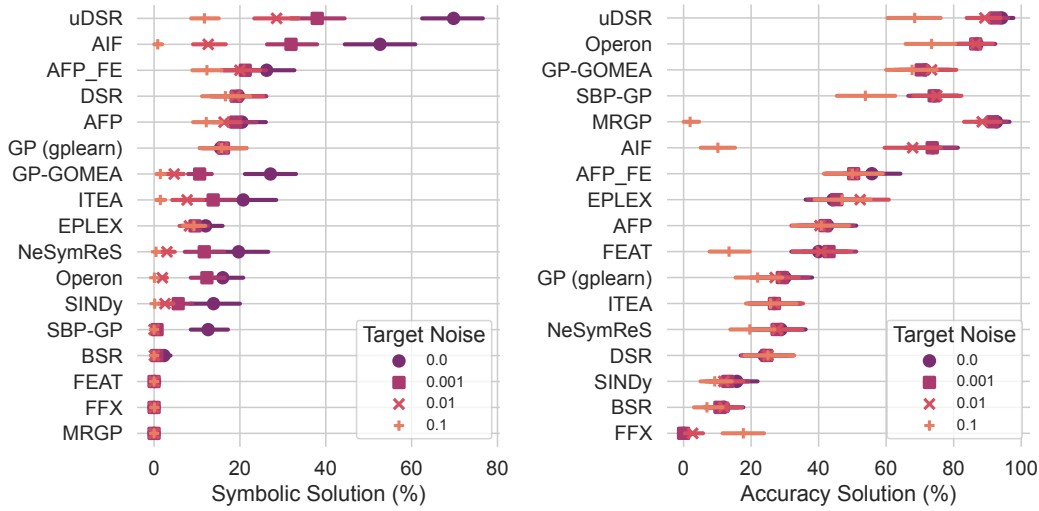

Figure 2: SRBench-generated comparisons of symbolic (left) and accuracy-based (right) solution rates for uDSR and 16 baseline SR methods, averaged across 130 ground-truth SR problems and four different noise levels.

in Table 3 of Appendix F. When possible, we reused the hyperparameters of the published methods most similar to each uDSR component.

We ran uDSR via SRBench on the 252 datasets with 10 random seeds each. Each run was performed for up to 2,000,000 expression evaluations per sub-problem, or until a 24 hr walltime limit was reached. For ground-truth problems, SRBench computes two metrics, each averaged across the 130 problems: (1) *symbolic solution* rate, based on symbolic equivalence using SymPy (Meurer et al., 2017) and (2) *accuracy solution* rate, defined as achieving an $R^2 > 0.999$ on held-out test data. For black-box problems, SRBench reports accuracy ($R^2$ on held-out test data) vs complexity (number of nodes in the SymPy-parsed expression tree).

**Performance results.** Figure 2 shows that uDSR outperforms all other 14 benchmarked methods both in symbolic solution (by a large margin) and accuracy solution rates. Figure 3 shows that uDSR falls on the Pareto frontier, alongside Operon (Kommenda et al., 2020) (at higher accuracy and complexity), and DSR and SINDy (at lower accuracy and complexity).

Notably, the previously published SRBench results had no clear winner across the three categories of (1) symbolic solution, (2) accuracy solution, and (3) Pareto efficiency. The top performers for symbolic solution rate (namely, AIF) differed from those for accuracy solution rate (namely, Operon and MRGP). Only Operon appeared as a top performer in two categories. However, in this work, we see that uDSR is highest in all three categories.

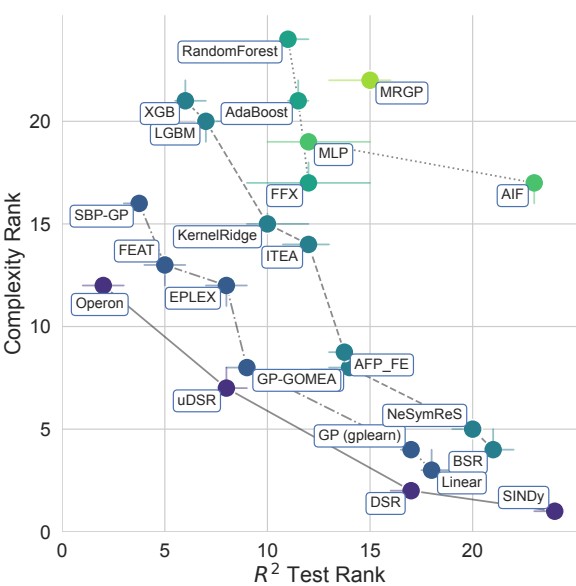

Figure 3: SRBench-generated comparison of test $R^2$ (on held-out test data) vs complexity for uDSR and 22 baseline regression/SR methods, averaged across 122 black-box SR problems.

## 6 Ablations and Discussion

**Ablations.** Since uDSR integrates many disparate components, it is critical to assess the relative contributions of each component in various contexts. To identify synergies, anti-synergies, and

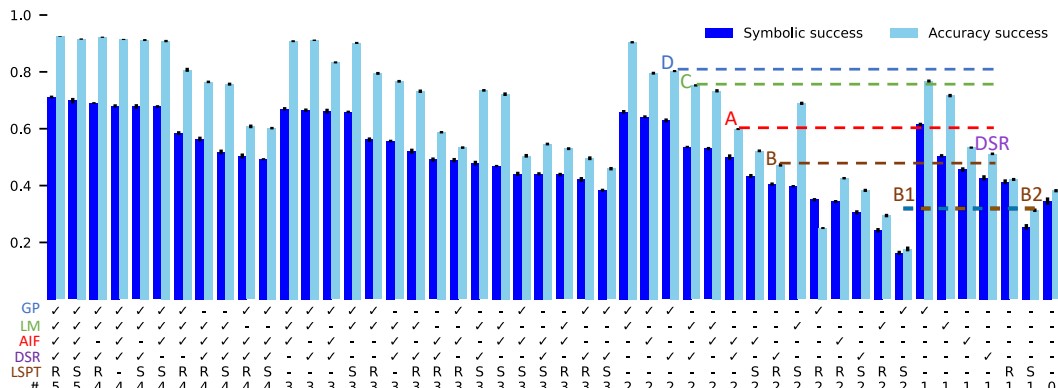

Figure 4: Combinatorial ablations of uDSR components on 130 SRBench ground-truth problems. Indicators below bars denote which components are enabled for each ablation. ✓: Component enabled. -: Component disabled. S: Pre-training enabled using SL. R: Pre-training enabled using RL. #: Number of enabled components. Error bars represent standard error across 10 random seeds per problem. Colored labels are referred to in the main text. Ablation 'D' is equivalent to Mundhenk et al. (2021). Ablation 'DSR' is equivalent to Landajuela et al. (2021b).

diminishing returns among the integrated methods, we perform a large-scale study of *all combinations* of the five integrated methods (standard "leave-one-out" style ablations in which we begin with the full uDSR and ablate one component at a time do not provide this full picture). There are 48 total combinations: AIF ∈ [on, off] × GP ∈ [on, off] × LINEAR ∈ [on, off] × pre-training ∈ [SL, RL, off] × DSR ∈ [on, off]. We define ablating each component as follows: For AIF, "off" means that we do not perform recursive problem-simplification; only the root problem is considered. For GP, "off" means we do not perform inner-loop GP algorithms between batches. For LINEAR, "off" means we exclude the LINEAR token from the library. For pre-training, SL and RL refer to whether we use the model pre-trained using SL or RL; "off" means we do not use any pre-trained model, and the controller architecture does not include the set transformer component. For DSR, "off" means the learning rate for the neural network is set to zero; notably, the controller is still used for sampling, including all priors and constraints. For each ablation, we run on the 130 ground-truth SRBench problems with 10 random seeds each ($48 \times 130 \times 10 = 62,400$ total runs), using a maximum of 500,000 expression evaluations per sub-problem.

Figure 4 shows the symbolic solution rate and accuracy solution rate computed by SRBench, with labels for select ablations we refer to in the subsequent discussion. Generally, performance increases with the number of enabled components, $N$. For a fixed $N$, GP and LM tend to yield the largest marginal performance gains. Refer to Figure 9 in Appendix G for a visualization of the marginal performance gains yielded by each component.

**Recommendations.** The full uDSR algorithm, using all components and RL pre-training, achieves the best symbolic and accuracy success on SRBench. Nonetheless, we believe that the choice of SR algorithm should be guided by use case. Thus, based on the extensive ablation studies presented above, we provide our recommendations for the five integrated solution strategies.

AIF's recursive simplifications have the attractive property of providing *strict improvement* over the base SR algorithm, since the root sub-problem in the sub-problem tree is simply the original problem; sub-problems further down the tree only have the ability to improve the end result. This property comes at the expense of incurring a roughly $m$-fold increase in computational cost, where $m$ is the number of identified sub-problems. Empirically, our ablations confirm the strict improvement (e.g., compare 'A' with 'DSR' in Figure 4), though performance gains exhibit diminishing returns as the number of integrated methods increases. Thus, we suggest inclusion of AIF for use cases in which computational cost is not an important bottleneck. Finally, we note that we only considered AIF without its dimensionality analysis component, as this is really a pre-processing step orthogonal to the SR algorithm. For real-world problems with physical units, we highly recommend that users first perform dimensional analysis on their SR problem.

The best results were obtained using RL pre-training with all other components enabled. However, in general, LSPT across the ablations yielded mixed results. For example, the only significant anti-synergy among all components occurs between SL pre-training and GP (compare 'B1' and 'B2'

in Figure 4). To gain further insight into which use cases may benefit from LSPT, we inspected the initial rewards for each pre-training modality *before* fine-tuning. As shown in Figure 6 of Appendix D, both SL and RL pre-trained models yield higher reward than randomly initialized models. This is especially the case for RL pre-training, where the traces of the initial expressions (see Figure 8 of Appendix D) appear very close to the real data. Thus, pre-training can be desirable for use cases with very low budgets (e.g., zero-shot learning), a point reinforced empirically by Biggio et al. (2021). However, as shown in Figure 7 of Appendix D, pre-trained models may have tendencies to become too "sharp" (i.e., exhibiting low entropy), resulting in reduced exploration that can hinder performance during fine-tuning. We conclude that to be effective in combination with other solution strategies in SR, pre-trained models must strike the delicate balance between high initial reward and sufficient initial diversity.

Both the LINEAR token and the inclusion of our GP component almost always provide large improvements in both symbolic solution and accuracy solution (e.g., compare 'C' and 'D' with 'DSR' in Figure 4). They also do not incur significant additional computational cost given a fixed budget of expressions (see Table 4 of Appendix H). Thus, we highly recommend their inclusion for most use cases. A limitation of the LINEAR token, and to a lesser extent the use of GP, is that it is inefficient in improving accuracy relative to its complexity, which we show in Figure 10 of Appendix I. Thus, users may not want LINEAR or GP for use cases in which they seek a low-complexity expression.

## 7 Conclusion

We introduce a modular framework for SR, in which we carefully integrate five different SR solution strategies in an attempt to maximize the benefits of each and cover each other's weaknesses. Our unification strategy focuses on *abstraction* of the constituent strategies: using DSR as a baseline, we abstract AIF as an algorithmic wrapper, LSPT as a pre-training step, GP as an inner-optimization loop, and LM as a single optimizable token. Using the SRBench pipeline for benchmarking SR algorithms, uDSR demonstrates state-of-the-art performance across 252 benchmark problems. For future work, we leave open the questions of using more advanced modules for GP, such as GP-GOMEA (Virgolin et al., 2017), or controller architectures (such as transformers as alternatives to the RNN), as well as the possible benefits from hyperparameter tuning individual modules. We hope that future work in SR considers additional strategies for hybridizing and/or abstracting SR solution strategies.

## Acknowledgments and Disclosure of Funding

We thank Livermore Computing at Lawrence Livermore National Laboratory (LLNL) for the computational resources that enabled this work. Funding was provided by the LLNL Laboratory Directed Research and Development project 19-DR-003. We thank the Computational Engineering Directorate and the Data Science Institute at LLNL for additional support. This work was performed under the auspices of the U.S. Department of Energy by LLNL under contract DE-AC52-07NA27344. LLNL-CONF-835375.

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
