# OpenReview forum: "A Unified Framework for Deep Symbolic Regression"
_NeurIPS.cc/2022/Conference — NeurIPS 2022 Accept_

### Official Review · Reviewer_xSLH · 2022-07-07

**Rating:** 7
**Confidence:** 4
**Soundness:** 3 good
**Presentation:** 3 good
**Contribution:** 3 good

**Summary:**

This work proposes a new symbolic regression method that unifies 5 existing symbolic regression methods and assesses the approach using SRBench a recently proposed benchmark framework for symbolic regression. It also discusses contribution of each component to the unified framework through  comprehensive ablation studies (48 combinations of the components used in the framework) and provides some recommendations regarding the component choice based on the ablation studies.


**Questions:**

- Use of the inverse function (L223) does not look like a good option in practice, but how can the author assure that it can be generalized and applied to real-world problems? e.g., what if $y - \sin(x_1) \leq 0$ in $\bar{f}^{-1}(y) = \log(y - \sin(x_1))$
- How many of the 252 datasets could the proposed method not finish the inference within the 24 hours walltime?
- How did this study obtain the results of the 14 baseline methods? If it just refers to the SRBench paper and shows the copy of the results, the authors should clearly describe so in the paper and explain whether or not the noise-injected datasets (Fig. 2) are identical to those in the SRBench paper. If the authors reran the baselines (, which would be preferred), the authors should mention so and explain computing resource used for the baselines.
- “the highest Pareto ranking,” (L302) should be clarified in the paper.
- If the authors can provide (pre)training time and inference time for each configuration in Fig. 4, it will be interesting and support some of the statements such as L351-352.
- “For real-world problems with physical units,” (L354-355) what datasets in the authors’ mind for instance? i.e., If none of the datasets used in this study is categorized into it, how did the authors find the dimensional analysis on the SR problem recommended?
- Table 2 makes the reviewer think if SINDy is contributing to the framework as its only one property is already covered by AI Feynman.

# Misc
- Figure 1 should be placed around Section 4, where the figure is first referred.
- **NSRS** should be replaced with **NeSymReS** to respect the name used in the original study (Biggio et al., 2021)
- “where F is a nonlinear function subject to certain constraints (see below)” Please specify where it can be found.
- The repeated use of “thesis” in this paper sounds weird. Maybe it means either “statement”, “claim”, or “argument”?

**Limitations:**

While the authors state in the checklist that several limitations are discussed in Section 6, the reviewer found only one limitation explicitly mentioned (L374-375). The reviewer did not find any potential negative societal impact of this work discussed in this work.
As explained in **Weaknesses** above, the main limitations of this work are the evaluation metrics (not well explained or justified either) and datasets (looks overlapping the ranges of values in generated datasets used for pretraining).


**Strengths And Weaknesses:**

Overall, this study provides interesting findings of symbolic regression, and the reviewer found that the study may be beneficial for the community while there are some things to be addressed/clarified.

# Strengths
1. From the motivations described and ablation studies in the paper, the proposed, unified framework for deep symbolic regression seems carefully designed, and it is not like just a combination of random existing methods.
2. The ablation study to discuss which component of the proposed framework contributes to the improvement seems very comprehensive. The community may also learn from recommendations based on the ablation study.
3. Although the reviewer has some concerns in the SRBench-based evaluation in this study (see **Weaknesses**), it seems that the proposed framework provides improved performance for datasets used in SRBench, comparing to existing symbolic regression methods.


# Weaknesses
1. Some of the evaluation metrics used in this study (symbolic/accuracy solution rate, $R^2$, accuracy solution) should be clarified and justified in the paper (i.e., the paper should be self-contained). While these metrics are used in SRBench, the reviewer is concerned how essential for SR each of these metrics is.
2. It seems that NeSymReS (referred to as NSRS in this study) and SINDy are both missed in evaluation as baselines while those are components of the proposed method and the baseline results for the other components (AIF, DSR, and GP) are provided.
3. The reviewer is afraid that the choices of ranges of variables in the instance generation for pretraining are very heuristic and potentially leaked from the datasets used for evaluation in this study. According to Algorithm 3 in Appendix, the ranges of sampled values for variables seem overlapping those of Feynman datasets used in this study e.g., most of them are within range of 1 to 10 (See FeynmanEquations.csv and BonusEquations.csv available at [the original dataset provider](https://space.mit.edu/home/tegmark/aifeynman.html))

---

> ### Author Response · Authors · 2022-08-02
> **Response to Reviewer xSLH (1 of 2)**
>
> **Evaluation metrics.** Regarding defining the SRBench evaluation metrics for "symbolic recovery" and "accuracy solution," these definitions were provided in the text: "symbolic recovery, based on symbolic equivalence using SymPy" (L293) and "accuracy, defined as achieving an $R^2 > 0.999$ on held-out test data" (L294). However, we now observe that our usage of these two metrics varied slightly throughout the text (e.g. "solution rate" and "recovery rate" were used interchangeably); we have edited the paper to streamline these two metrics as "symbolic recovery" and "accuracy solution." $R^2$ refers to the coefficient of determination.
>
> Regarding justification for these metrics, symbolic recovery is a critical metric for SR in assessing its ability to exactly rediscover governing equations. A challenge with this metric is that definitions of recovery differ across works. SRBench (along with AIF) uses SymPy to symbolically assess symbolic recovery, which is highly stringent compared to methods that base recovery on an arbitrary error threshold.
>
> Having an error-based metric is also important to SR, especially for black-box problems in which recovery is not possible. $R^2$ in particular is justified because it is normalized across datasets, and is thus reasonable to average $R^2$ scores across different SR problems. $R^2$ is also scale invariant (unlike mean square error or mean absolute error). We note that error-based metrics are more valuable when paired with complexity-based metrics and compared in a Pareto sense. This is why Figure 3 is also critical in assessing uDSR's overall performance.
>
> Finally, taking a step back, a major value of the existence of SRBench is for establishing conventions for performance metrics in SR, which were previously painfully lacking in the community. Thus, even if some SRBench choices are not perfect (e.g., we believe that some notion of "accuracy solution" is not critical to SR and that SRBench's $R^2 > 0.999$ cutoff is arbitrary), the SRBench metrics are being rapidly adopted (already being used in several SR works, as mentioned in L144 - 146), which enables direct literature-to-literature comparison.
>
> **Including NeSymReS and SINDy as baselines.** While NeSymReS and SINDy were initially excluded from our SRBench-generated plots (Figures 2 and 3) as they were not originally benchmarked by SRBench, we agree that including them would be valuable to allow users to compare all 5 individual baselines against uDSR. Thus, we re-added these baselines into Figures 2 and 3. Not surprisingly, SINDy performs poorly in Figure 2, as most benchmarks are not linear combinations of simple basis functions, though its sparse solutions place it at the low-complexity extreme of the Pareto frontier for black-box problems (Figure 3). NeSymReS performs similarly to our NeSymReS-only ablations.
>
> **Ranges of variables used for pre-training.** The domain extrema for the variables used to generate pre-training data (randomly sampled between -10 and 10) were reused from the original NeSymReS publication without any consideration of test problems. As mentioned in L288 - 289, this follows our design philosophy of reusing hyperparameters from the five integrated methods wherever possible. Notably, the NeSymReS paper did not compare against any of the SRBench problems in this work, so we do not believe there was risk of test information leak—which we agree would be a major concern. Moreover, it looks like none of the Feynman benchmark problems include any negative values, yet 75% of our generated pre-training problems exhibit a domain spanning negative values. The Strogatz benchmarks also include data well outside the domain extrema considered during pre-training: for example, the [bacres1](https://epistasislab.github.io/pmlb/profile/strogatz_bacres1.html) benchmark has an input variable as high as 54.4.
>
> (continued below)

---

> > ### Author Response · Authors · 2022-08-02
> > **Response to Reviewer xSLH (2 of 2)**
> >
> > (continued from above)
> >
> > Below we address each each point raised in the "Questions" and "Misc" sections.
> >
> > * The reviewer correctly makes the astute observation that the expression for $\bar{f}^{-1}(y)$ may be undefined for some values of $(X, y)$. In these cases, the evaluation of the LINEAR token "aborts" and returns a default value of LINEAR $= 1$. We will clarify this detail in the text. We note that if the functional form is correct (e.g. the traversal $\tau =$ [exp, LINEAR] when the ground-truth expression is $e^{x_1^2 + 0.5x_2}$), then, by construction, it will never be the case that $\bar{f}^{-1}(y)$ yields undefined values.
> > * Of the >70,000 independent trials of uDSR spanning this work, around 1\% reached the 24 hr limit. Across the 252 total SRBench problems (130 ground-truth, 122 black-box), timeouts only ever occurred on 14. Since the computational bottleneck is evaluating the candidate expressions, timeouts were concentrated on problems with larger datasets; for example, the few black-box problems that included ~1M $(X, y)$ pairs.
> > * As mentioned in L283 - 285, the results of the 14 SRBench baselines were taken directly from the curated, open-source SRBench data. As SRBench aims to be reproducible and facilitate direct comparison, the noise-injected studies are identical (i.e., it is the exact same noise values per problem being added to all baselines and to uDSR).
> > * The phrase "achieves the highest Pareto ranking" should simply read "falls on the Pareto frontier." We have corrected this mistake.
> > * Regarding inference runtimes of the 48 ablation studies, these are already provided in Table 5 in the Appendix I. We opted for a table instead of barchart here to allow users to more quantitatively compare runtimes with different components to better inform decisions for their own use cases. Pre-training times were also provided in L696 - 697: 8 days on a single GPU for each of RL and SL pre-training. Appendix J provides additional computing infrastructure details.
> > * Many real-world datasets have associated physical units. Given these, the dimensionality analysis (DA) module of AIF can be exploited to greatly simplify the SR problem. These problems were not considered in this study because DA can be performed orthogonally to any SR method. Our recommendation to use DA when applicable stems from the dramatic performance increase when adding DA as shown by AI Feynman [1].
> > * Regarding the SINDy property in Table 2 already being covered by AIF, the difference is that SINDy induces sparsity in their linear models. We have updated that column name to "Sparse linear FPO" and removed the corresponding checkmark for AIF.
> > * We believe the placement of Figure 1, illustrating the overall pipeline, can provide a useful aesthetic on the front page. We have found it common to include an summary figure or result on the front page even when referenced much later, for example [1] and [2]. Nevertheless, we have also added a reference to our Figure 1 in the Introduction.
> > * We have edited all instances of NSRS with the authors' original acronym, NeSymReS.
> > * Regarding the "certain constraints (see below)" phrase regarding the LINEAR token, these constraints are detailed in L236 - 240.
> > * The two instances of the word "thesis" were replaced with "claim."
> > * While we only used the word "limitation" once when discussing limitations, Section 6 discussed several other limitations. L356 - 369 discusses the tendency of pre-trained models to hinder test-time performance via reduced exploration. L347 - 349 notes that the benefit of AIF's strict improvement comes at the extra cost of a fold-increase in computation.
> >
> > [1] Udrescu et al., 2020. AI Feynman 2.0: Pareto-optimal symbolic regression exploiting graph modularity. NeurIPS 2020.
> >
> > [2] Yu et al., 2020. Evaluating the search phase of neural architecture search. ICLR 2020.

---

> > > ### Comment · Reviewer_xSLH · 2022-08-07
> > > **Thank you for the clarifications**
> > >
> > > The reviewer thanks the authors for the clarifications.
> > > Some of the main concerns are resolved by the authors' response (partly because of the additional baseline experiments the reviewer requested), thus the reviewer improved the rating.
> > >
> > > > The domain extrema for the variables used to generate pre-training data (randomly sampled between -10 and 10) were reused from the original NeSymReS publication without any consideration of test problems.
> > >
> > > According to [FeynmanEquations.csv](https://space.mit.edu/home/tegmark/aifeynman.html) for the Feynman Symbolic Regression Database, the ranges of sampled values are quite narrow, and **most of them fits in range of 1 to 5**.
> > > Regardless of whether or not the original NeSymReS work used the range of -10 to 10 with some intention, the tested datasets seem to fit within the sub-ranges of the values used for pretraining.
> > > **It could be said due to the limitation of the Feynman Symbolic Regression Database that is a key dataset used as part of SRBench, and this study is heavily dependent on the benchmark.**
> > > It would be great if the authors could show a quantitative comparison between the ranges of values used in the tested datasets and the dataset used for pretraining.
> > >
> > > Lastly, the reviewer would expect the authors to reflect their response to the revision (and/or supplementary material if space is limited) and kindly provide the pointers to the updates in the revision and/or supplementary material with the line numbers.

---

> > > > ### Author Response · Authors · 2022-08-08
> > > > **Thank you for the continued discussion (1 of 2)**
> > > >
> > > > We greatly thank the reviewer for considering our replies and following up with continued valued discussion.
> > > >
> > > > **Ranges of input variables.** In general, test performance always depends on how well the test data distribution matches that used for pre-training. In this work, we strongly agree with the reviewer that input variable ranges are an important aspect of the data distribution. Notably, test problems where input variables values fall outside the ranges considered during pre-training result in extrapolation.
> > > >
> > > > We make several points regarding ranges of input variables:
> > > >
> > > > 1 - **The containment of the Feynman problem ranges does not dominate our pre-training problems.** Recall that domain extrema $[a, b]$ are *sampled from* [-10, 10] (L8 - 10 of Algorithm 3). Under this sampling scheme, 36.5% of pre-training problems *completely exclude* (do not allow a single data point within) the $[1, 5]$ range (i.e. $[a, b] \subset [-10, 1]\cup [5, 10]$); 27.5% span the whole [1, 5] range (i.e. $[1, 5] \subset [a, b]$); the remaining 36% partially overlap the [1, 5] range. Only 4% have extrema that fall completely inside [1, 5], i.e. $[a, b]\subset [1, 5]$.
> > > >
> > > > We added the **requested quantitative comparison in Figure 13**, which compares input variable ranges for pre-train, Feynman, Strogatz, and black-box problems.
> > > >
> > > > 2 - **It would have been very strange to design a pre-training dataset generator that does not produce input variables spanning the range [1, 5].** This range is near 0, and it is reasonable for any regression dataset generator to span 0. It would be nonsensical (without prior knowledge of test problems) to pre-train on data exclusively on a range like $[-100, -50]$, $[10, 100]$, or $[-10, 1] \cup [5, 10]$. It is difficult to imagine designing a pre-training dataset generator that does not generate a vast amount of data in the range [1, 5].
> > > >
> > > > 3 - **uDSR still outperforms all baselines on the SRBench ground-truth problems whose input ranges are not a subset of [-10, 10].** There are six such extrapolation problems, each from Strogatz. We added Table 6 which shows the performance of uDSR and all baselines averaged across these six problems.
> > > >
> > > > Standalone NeSymReS achieves 0% in both metrics. This provides evidence that testing *without fine-tuning* on problems outside the pre-training input ranges can be challenging.
> > > >
> > > > uDSR achieves 100% in both metrics, outperforming all baselines in both symbolic (by an extremely large margin) and accuracy solution rates, despite test problems extrapolating outside pre-training ranges. This may be due to uDSR fine-tuning at test time (unlike NeSymReS) which may help escape poor initial solutions (L365 - 369), and/or the fact that uDSR uses other components that are agnostic to input variable ranges (e.g. AIF and GP).
> > > >
> > > > 4 - **Similarly, uDSR performs well on black-box problems, despite the fact that 47 of 120 black-box problems contain input variables exceeding [-10, 10].** This is seen in Figure 3, in which uDSR is on the Pareto frontier.
> > > >
> > > > 5 - **As far as SR problem suites go, SRBench shows large variety in input variable ranges.** While Feynman problems only span [0, 20] (with most spanning [1, 5]), Strogatz span [-4.4, 54.4] and black-box span [-7466.9, 7322564]. In contrast, the heavily used Nguyen suite spans only [-1, 4], the Livermore suite in Mundhenk et al., 2021 spans [-10, 10], and [Biggio test problems](https://github.com/SymposiumOrganization/NeuralSymbolicRegressionThatScales/blob/main/test_set/nc.csv) span [-10, 10].
> > > >
> > > > Thus, while we agree that our study depends heavily on SRBench (emphasized in L141), and that SRBench is by no means perfect, it is a very reasonable choice and one that is quickly achieving widespread adoption (L144 - 146).
> > > >
> > > > 6 - **uDSR does not need the pre-training component to be state-of-the-art on SRBench.** As discussed in Recommendations (L356 - 359) and Appendix G (L828 - 831), the contribution of the NeSymReS component to the overall performance of uDSR is *rather minimal*. The "uDSR minus NeSymReS" ablation (Figure 4) still outperforms all Figure 2 baselines in both metrics, yet has no notion of input range choices.
> > > >
> > > > Thus, while we agree with the reviewer that design choices for NeSymReS (namely, variable ranges) may have a large impact relative to test problem ranges, we do not believe uDSR or the study as a whole is heavily reliant on these ranges in the Feynman benchmark problems.
> > > >
> > > > 7 - Finally, this discussion hearkens back to our conclusion that the SR algorithm choices should be driven by *use case*. For pre-training, a major use case consideration is the extent to which the user believes that their SR problems of interest matches the distribution of problems considered during pre-training. We added Appendix M.3 (L980 - 997 & Table 6) to discuss this and the above results. We believe this is a useful, practical point for our Recommendations section, and we thank the reviewer for continued discussion on this topic.
> > > >
> > > > (continued below)

---

> > > > > ### Author Response · Authors · 2022-08-08
> > > > > **Thank you for the continued discussion (2 of 2)**
> > > > >
> > > > > (continued from above)
> > > > >
> > > > > **Incorporating rebuttal context in text.** Due to page limitations during the rebuttal period, and in order to not interfere with original line number references, our first revised submission included small modifications that did not affect line numbers (e.g. NSRS $\rightarrow$ NeSymReS), changes to captions (which do not affect line numbers), and additional content at the end of the document (Appendix K & L, spanning L873 - 952).
> > > > >
> > > > > Existing changes discussed with this reviewer:
> > > > > * Figures 2 and 3 now include the new baselines NeSymReS and SINDy.
> > > > > * Changed "achieves the highest Pareto ranking" to "falls on the Pareto frontier" (L301 - 302)
> > > > > * Changed "Linear FPO" to "Sparse linear FPO" and unchecked AIF from that property (Table 2)
> > > > > * Replaced every instance of "NSRS" with "NeSymReS" (throughout the paper)
> > > > > * Changed "thesis" to "claim" (L100 & Footnote 1)
> > > > > * Referenced Figure 1 in Introduction (L52)
> > > > > * Noted $\sim$1% rate of hitting 24 hr walltime limit (L872; new to latest revision)
> > > > >
> > > > > From other reviewers:
> > > > > * Noted Mundhenk et al., 2021 baseline in Figure 4 caption
> > > > > * Added Appendix K discussing and benchmarking differences between standalone and "only-one-component" ablations (L873 - 940 and Figure 11)
> > > > > * Added Appendix L illustrating arrival at a solution (L941 - 952 & Figure 12)
> > > > >
> > > > > A few reviewer discussion points below we wanted to include in the main text but cannot at this time due to page limitations
> > > > > during the rebuttal period. For now, we have discussed these in a new revision in Appendix M, which, if accepted, we will incorporate into the main text. These topics are:
> > > > >
> > > > > * Behavior of non-finite target data using SINDy (Appendix M.1: L954 - 963).
> > > > > * SRBench clarifications: explanations of performance metrics; SRBench baselines vs SINDy/NeSymReS added baselines (Appendix M.2: L964 - 979)
> > > > > * Topics related to input variable ranges for pre-training (L980 - 997)
> > > > >
> > > > > We thank the reviewer again for their time and excellent discussion regarding this work.

---

> > > > > > ### Comment · Reviewer_xSLH · 2022-08-10
> > > > > > **Thank you for the the additional clarifications and analysis**
> > > > > >
> > > > > > The reviewer thanks the authors again for the continuous effort on clarifications and additional analysis to defend the work.
> > > > > >
> > > > > > Most of the main concerns the reviewer raised are resolved, and including these discussions would even strengthen this work.
> > > > > > The reviewer is now more convinced and made the (probably) final edit in their review to update the score.

---

### Official Review · Reviewer_PqgV · 2022-07-09

**Rating:** 5
**Confidence:** 4
**Soundness:** 3 good
**Presentation:** 3 good
**Contribution:** 2 fair

**Summary:**

This paper provides a holistic review of symbolic regression approaches, and picked 5 most popular ones for algorithm amalgamation to construct a new symbolic regression algorithm. The new symbolic regression algorithm is claimed to maximize the benefits of each and cover each other’s weaknesses.


**Questions:**


Below are some questions and suggestion regarding algorithm description and evaluation.

A concrete example is missing. Though the authors showed example formulas in line 221 and introduced the benchmark, it is important for the symbolic regression paper to introduce at least one example distilled symbolic rule, so as to make it easier for the reader to link what the authors are describing to which parts in the final distilled algorithm. Especially for this hybridizing work, could the author examplify which part in the symbolic rule is generate by which sub-problem? Especially the CINDy part, as how the vector based algorithm connects to symbolic optimization remains unclear to me, and could be made clear if explained via examples.

However, it did not show the possible downside of hybridizing different algorithms. Algorithm combination in a chain shaped form is known demonstrate more variance and less bias, so will the combined algorithm underperform the one single algorithm? ("If certain single algorithm is good enough, why combining with the other worse algorithm won't downgrade it?")

Though the author explained what the "on" and "off" mean in line 320, it is still unclear how different is the hybridized algorithm will be from some original algorithm, if all other sub-algorithms are turned off. Based on the results in fig 4, it seems that one single algorithm achieved ~ 0.6, which seems a bit lower than the results reported in the original papers. The ablation study focus on checking the components of the proposed algorithm, while the critical concern regarding this algorithm amalgamation model is what if one choose single sub-algorithm, and how does it compare with the hybridyzed algorithm. It is encouraged that the author evaluate more explicitely on how does the hybridized compared to single ones from the explanation side, and evaluate the result they got compared to the originally reported ones.


The workflow of the proposed hybridyzation is clearly explained both in Algorithm 1 and Fig 1. But the integration of the strength of the NSRS remains unclear to me. It seems that the NSRS has been downgraded to only generate expression with one specific (X,Y) problem. Is the proposed algorithm still possible to generalize across new problems?


**Limitations:**

Based on my limited assessment, this work stays free of possible negative societal impact.

**Strengths And Weaknesses:**


This paper studies an important problem, symbolic regression. The algorithm does not contain novel parts, but it offer novel way for us to understand the symbolic regression algorith especially what is the core parts and how could we combine them. The writting of this paper is clear enough. This paper provides thorough ablation studies, which showed different combinatorial ablations of uDSR components.

However, certain perspectives remains to be improved, including demonstration, and results explanation. I've put the corresponding suggestions and questions in the following, and would raise my assessment if they are properly addressed.

---

> ### Author Response · Authors · 2022-08-02
> **Response to Reviewer PqgV (1 of 2)**
>
> We thank the reviewer for their time and useful feedback and suggestions. First, we appreciate that the reviewer notes the novelty in distilling the core parts of existing SR algorithms and how to best combine them; however, we would like to make a clarification regarding the phrase "while the algorithm does not contain novel parts". The integration of uDSR components required several novel components, including (1) a novel methodology for generalizing SINDy into a single token and (2) new theory and corresponding gradient estimate for the risk-seeking policy gradient in the context of pre-training on conditioned problem instances. We point the reviewer to the section "Technical novelty" in our response to Reviewer CaGz for more detailed descriptions of the novelty.
>
> **Concrete example.** Because the 5 algorithms work closely together (rather than as a "chained" or "pipeline" approach), it is not always possible to tease out exactly which parts of a symbolic expression were derived from which component. In particular, for NeSymReS, it is not possible to determine whether a specific expression was sampled due to the fact that the distribution had been pre-trained (versus beginning with a randomly-initialized distribution).
>
> However, we *can* retroactively tease out this information for most components. Because SINDy was abstracted into an individual token, it is straightforward to retroactively determine which part of the expression came from this component. Similarly, since AIF's recursive simplifications track individual sub-problems, it is possible to retroactively determine how individual sub-solutions form the final solution. For GP, we can identify the solution's parent expression(s) and the GP operation that resulted in the child solution.
>
> Thus, as suggested, we added Figure 12 (new in Appendix L), which provides an illustrative example of how different parts of a final solution may be attributed to different components. In particular, we include the matrices and vectors involved in the optimization of the LINEAR token (SINDy). We thank the reviewer for this suggestion, as it provides a clean visual representation of how the uDSR components can work together to form a solution.
>
> **Possible downsides of combining multiple algorithms.** In general, we agree with the reviewer that hybridizing multiple algorithms together is not guaranteed to improve performance, and may even degrade performance relative to a single algorithm.
>
> We first clarify that our five constituent algorithms were not simply "chained" or "pipelined" together; rather, they were carefully integrated into a cohesive yet modular framework. This is noted by Reviewer xSLH's "Strength 1." In particular, the integration strategy for several of our uDSR components was specifically designed to mitigate or eliminate the possibility of performance degradation. For example, the AIF component yields a theoretical *strict performance improvement* over uDSR configurations with AIF disabled (L346 - 348), which we empirically confirmed in L350 - 352. (The accompanying trade-off of increased computational cost was discussed in L348 - 350, with data provided in Table 5.) Similarly, our novel LINEAR token is a generalization over the original SINDy search space, which recovers SINDy for the simple traversal $\tau$ = [LINEAR].
>
> However, as the reviewer correctly points out, degradation is still possible in an integrated framework. The main uDSR component for which integration may degrade performance is NeSymReS, because fine-tuning from a pre-trained model may yield locally optimal solutions that could have been avoided if beginning from a randomly-initialized model. This was discussed in our Recommendations section in L366 - 370, where we noted the pros and cons of integrating NeSymReS.
>
> The topic of possible performance degradation (referred to as "anti-synergies" in our text) was more closely examined in Figure 9, which plots the marginal performance improvements obtained by including each uDSR component across the 48 configurations. We see that 4 of the 5 components exhibit marginal performance gains in most settings, with strictly positive interquartile ranges. In contrast, NeSymReS can often reduce marginal performance, a topic discussed in L357 - 360, again in L828 - 830, and investigated more deeply in Figures 6 - 8.
>
> (continued below)

---

> > ### Author Response · Authors · 2022-08-02
> > **Response to Reviewer PqgV (2 of 2)**
> >
> > (continued from above)
> >
> > **Original methods vs uDSR components.** The reviewer is correct that there are methodological differences between the 5 original methods (as published, standalone SR algorithms) and the 5 corresponding uDSR components. These differences are detailed in Methods; however, to provide a more localized summary of differences, we summarize them in Appendix K (new).
> >
> > Regarding performance comparison between the hybrid algorithm (uDSR) and the 5 original methods, most of these can already be found in Figure 2, which includes the as-published versions of AIF, DSR, and GP (denoted "gplearn"). SINDy and NeSymReS (as published) were not included as part of SRBench's original set of baselines, and were thus not included in Figures 2 or 3. However, as requested, we ran SINDy and NeSymReS (as published) via the SRBench pipeline and added the results to Figures 2 and 3. Not surprisingly, SINDy performs poorly in Figure 2, as most benchmarks are not linear combinations of simple basis functions; NeSymReS performs similarly to our NeSymReS-only ablations. Thus, Figures 2 and 3 now includes all 5 comparisons between uDSR and the 5 as-published methods (along with various other SRBench-benchmarked methods). We thank the reviewer for this suggestion.
> >
> > The reviewer also correctly points out that there can be performance differences between the as-published methods and the corresponding uDSR "only-one-component" ablations (i.e., ablations in which only 1 of the 5 uDSR components are enabled); for example, AIF as a standalone algorithm achieves an accuracy solution rate of >0.7 (Figure 2), whereas the AIF-only uDSR ablation achieves <0.6 (Figure 4). (In this case, the difference is due to AIF sampling via brute force + polynomial fit, whereas the uDSR ablation samples from a randomly-initialized controller; AIF as published also uses a token set that is tailored to solving the Feynman benchmark problems.)
> >
> > To facilitate direct, side-by-side comparisons between the 5 as-published algorithms and the 5 corresponding uDSR "only-one-component" ablations, we added Figure 11 (new in Appendix K). Note that this figure does not contain any new data relative to existing figures; however, it enables baseline-to-ablation comparisons without requiring readers to compare across Figure 2 (performance comparisons to standalone baselines) and Figure 4 (ablations of uDSR components). In short, AIF is the only standalone algorithm that outperforms the corresponding uDSR "only-one-component" ablation. Further discussion into these performance differences is found in Appendix K.2.
> >
> > **Integration with NeSymReS.** We reassure the reviewer that our pre-trained models using the NeSymReS component indeed generalize to many new SR problems. We pre-trained a *single* model on ~15 million expressions, which was then reused as a starting point for subsequent fine-tuning on each test problem. (To span all ablation studies, we pre-trained a total of 4 models: 2 with RL and 2 with SL, each with/without the LINEAR token.)
> >
> > The difference between our use of the original NeSymReS work and our NeSymReS component is *how* inference is performed at test time. We first clarify the similarities. During pre-training, **both** NeSymReS (as published) and our component pre-train a distribution over expressions on many SR problems, using an encoder to condition each $(X, y)$ dataset. At inference/test time, **both** NeSymReS (as published) and our component generate expressions for a single $(X, y)$ dataset by conditioning the distribution on this data. The key difference is *how* expressions are generated at test time. NeSymReS as published performs *beam search* on the conditioned distribution, which identifies high-likelihood expressions under the pre-trained model, which are then evaluated for fitness. In contrast, our NeSymReS component generates expressions by *fine-tuning* the conditioned distribution using the remaining four uDSR components.
> >
> > Lastly, we note that the computational bottleneck of uDSR is computing error between the sampled expression and the $(X, y)$ problem data: about 90\% of compute is spent evaluating expressions. Thus, computational cost of the original NeSymReS inference (beam search) is approximately the same as our inference (fine-tuning) per candidate expression.

---

### Official Review · Reviewer_CaGz · 2022-07-12

**Rating:** 5
**Confidence:** 4
**Soundness:** 2 fair
**Presentation:** 3 good
**Contribution:** 2 fair

**Summary:**

This paper tackles a symbolic regression (SR). SR is a task to explain observed data using mathematical expressions. The authors state that there are five individual methods and the proposed method combines them. The experimental results on SRBench, a benchmark for SR, demonstrate that the proposed method performs better than the existing methods.

**Questions:**

The reviewer would like the authors to solve the three weaknesses above.


**Limitations:**

There are some descriptions of technical limitations, not societal ones.


**Strengths And Weaknesses:**

#### Strengths
- Combining the individual methods for SR is interesting. The view is also helpful in understanding the current trend of the SR methods.
- The overall architecture and experimental procedure are well described in the main text and supplementary material.
- The experimental result shows that the proposed method, uDSR achieves better performance than the existing ones.

#### Weaknesses
- Since the proposed method is a combination of the existing methods, its technical novelty is limited.
- There is another method that combines several types of SR methods [a] that is shown to outperform current state-of-the-art methods. Some comparisons to uDSR were necessary.
- Figure 4 is great in terms of ablation study, but it seems that the combination of all five modules achieves statistically significant improvements over the other variations.

[a] Mundhenk et al. Symbolic Regression via Neural-Guided Genetic Programming Population Seeding. NeurIPS, 2021.

---

> ### Author Response · Authors · 2022-08-02
> **Response to Reviewer CaGz**
>
> We thank the reviewer for their time in reviewing our work. We believe the responses below should alleviate concerns of the stated weaknesses.
>
> **Technical novelty.** We do not believe that the nature of this work (a hybridization of existing methods) limits its technical novelty. We recognize that it may be easy to view a hybridization algorithm as having little novelty on the surface. However, we re-emphasize that in this work we hybridize the *key capability* of each of the 5 constituent methods; we do not simply "chain" these works together or use them in their original published form. Rather, integration into a single coherent framework required careful redesign, modifications, development of new theory (for NeSymReS), and/or completely new methods (for SINDy) relative to the original works. This important distinction is discussed in Section 4 (L155 - 163), clarified in Section 6 (L327 - 334), and visualized in Table 1.
>
> In particular, we stress that the LINEAR token is highly novel, and we believe this token alone to be a substantial contribution to the field of symbolic regression (SR). The creation of this token involves a novel methodology involving partial execution, lexical closures, and successive tree inversion (L211 - 230). The result is a (non-linear) *generalization* of the SINDy search space (SINDy itself becomes the trivial case of the single-node traversal $\tau$ = [LINEAR]), which still retains the key capability of obtaining computationally expedient sparse linear sub-models. Thus, the technical connection between the original SINDy algorithm and our LINEAR token is actually quite weak/nominal; while the LINEAR token was loosely based on (inspired by) the idea of sparse linear modeling used in SINDy, its technical novelty is largely unrelated.
>
> Another major technical novelty is the theoretical contribution and corresponding Monte Carlo gradient estimate required to adapt the risk-seeking policy gradient to the pre-training setting in which the controller is conditioned on each problem instance (L271 - 275 and Appendix D.4 in the Supplementary Material). NeSymReS as published does not consider RL-based training; thus, while our contribution is related to the key capability of NeSymReS that inspired our work (large-scale pre-training), our technical novelty is orthogonal.  For instance, in contrast with NeSymReS, this method can be used to pre-train systems for discrete search without access to ground-truth labels (i.e. on black-box SR problems).
>
> **Comparison to Mundhenk et al., 2021.** We agree that the Mundhenk et al., 2021 is an important baseline. This baseline is already explicitly included as one of the 48 ablations. Specifically, it corresponds to the ablation with only DSR and GP enabled, labeled 'D' in Figure 4. The correspondence between this ablation and Mundhenk et al., 2021 is exact, as we used the same GP integration strategy (L247 - 253), same GP hyperparameters (Table 3), and same open-source code (L861 - 863).
>
> We agree it was an oversight to not make this connection explicit to the reader; we have now clarified this in the Figure 4 caption by adding "Ablation 'D' is equivalent to Mundhenk et al., 2021." Similarly, we have clarified in the caption that the DSR-only ablation exactly corresponds to DSR as published [1]. Note that none of the other 46 ablations exactly correspond to originally published methods.
>
> **Performance improvement with all modules activated.** *Remark*: We assume the reviewer meant "...the combination of all five modules DOES NOT achieve statistically significant improvements."
>
> We performed statistical significance tests between the most performant ablation $X_\textrm{best}$ (in which all components are activated, using supervised learning pre-training) against each of the other 47 ablations $X_i$. Specifically, we tested the null hypothesis that "the means of $X_\textrm{best}$ and $X_i$ are equal." Of these 47 significance tests, only 3 yielded $p$-values greater than 0.01. These three ablations each had 4 of the 5 components enabled. This reinforces our claim that while there are diminishing returns when combining the 5 components together, there is still a very strong trend that they improve performance (further detailed in Appendix G and Figure 9). More importantly, our overall message is that use of each component should be driven by use case, as discussed in our Recommendations section (L341 - 376).
>
> Lastly, we thank the reviewer for noting the value of the extensive ablation studies. We believe that performing such combinatorial ablation studies is lacking in the field of machine learning; even the seminal Rainbow paper [2] does not provide fully combinatorial ablations, only "all-but-one-component" and "only-one-component" ablations.
>
> [1] Landajuela et al., 2021. Improving exploration in policy gradient search. Math-AI Workshop, ICLR 2021.
>
> [2] Hessel et al, 2018. Rainbow: Combining improvements in deep reinforcement learning. AAAI 2018.

---

### Meta-Review · Area_Chair_ENPy · 2022-08-26

**Recommendation:** Accept
**Confidence:** Certain

**Metareview:**

The paper presents a novel deep symbolic regression approach that is a hybridization of existing methods, showing state-of-the-art performance on the SRBench. Let me stress that being a hybrid is no reason to reject a paper as creating hybrids can be a very creative contribution. And for me this is the case here. the hybrid is not just a "mixture" but actually a very creative rewiring of components of the underlying approaches. This is also supported by the ablation study. Moreover, several of the
issued raised were clarified well in the rebuttal.  BTW, the authors may also want to cite other approaches for equation discovery, see e.g.

Jure Brence, Ljupco Todorovski, Saso Dzeroski:
Probabilistic grammars for equation discovery.
Knowl. Based Syst. 224: 107077 (2021)

Will Bridewell, Pat Langley, Ljupco Todorovski, Saso Dzeroski:
Inductive process modeling. Mach. Learn. 71(1): 1-32 (2008)

But this only for making the paper more self-complete.

**Award:**

No

---

### Decision · Program_Chairs · 2022-09-14

Accept